# Dural Metastases of Advanced Prostate Cancer Detected by ^18^F-Fluorocholine

**DOI:** 10.3390/diagnostics10060385

**Published:** 2020-06-08

**Authors:** Mauro Morassi, Mattia Bonacina, Claudio Bnà, Alberto Zaniboni, Giordano Savelli

**Affiliations:** 1Division of Radiology, Fondazione Poliambulanza Istituto Ospedaliero, 25124 Brescia, Italy; mauro.morassi@poliambulanza.it (M.M.); claudio.bna@poliambulanza.it (C.B.); 2Division of Nuclear Medicine, Fondazione Poliambulanza Istituto Ospedaliero, 25124 Brescia, Italy; mattia.bonacina@poliambulanza.it; 3Department of Medical Oncology, Fondazione Poliambulanza Istituto Ospedaliero, 25124 Brescia, Italy; alberto.zaniboni@poliambulanza.it

**Keywords:** ^18^F-fluorocholine, prostate cancer, computed tomography, dural metastases, subdural hematomas

## Abstract

Prostate cancer with extensive dural metastases is very rare, with only few cases described in the literature. We report one such case of a 74-year-old man with advanced prostate cancer, and in relatively good clinical condition. The patient returned with complaints of headache and diplopia. Fluorocholine (^18^F) chloride (^18^F-FCH) is an analog of choline in which a hydrogen atom has been replaced by fluorine (^18^F). After crossing the cell membrane by a carrier-mediated mechanism, choline is phosphorylated by choline kinase to produce phosphorylcholine. ^18^F-FCH positron emission tomography–computed tomography (PET/CT) is widely used to stage and restage patients affected by prostate cancer with good sensitivity. ^18^F-FCH PET/CT showed disease progression with the onset of multiple skull lesions. Numerous suspicious dural hypermetabolic lesions indicating neoplastic involvement were detected along the fronto-parietal convexities, in the left fronto-orbital region and right lateral wall of the orbit, concerning for metastases in these regions. A contrast-enhanced computed tomography (CECT) scan was performed which showed corresponding enhancing tissue which correlated with the PET findings. The final imaging diagnosis was osteo-dural metastases from prostate cancer associated with poor outcome. Awareness of this pattern of metastases may be of clinical relevance in order to avoid unnecessary invasive diagnostic procedures in groups of patients with a dismal prognosis.

**Figure 1 diagnostics-10-00385-f001:**
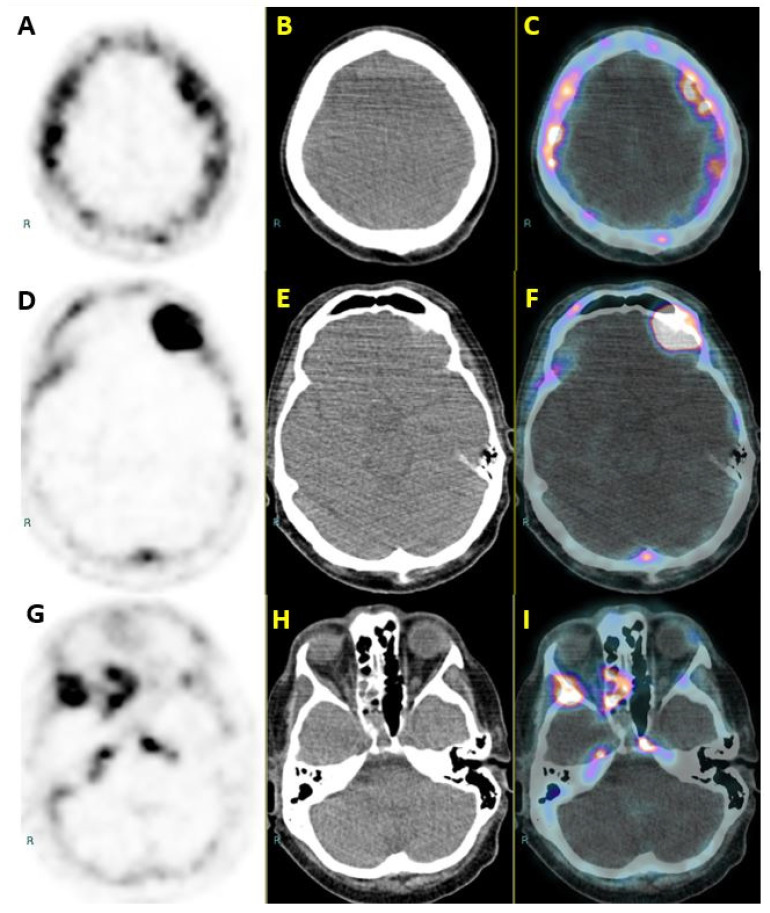
A 74-year-old man was referred to our center for a ^18^F-fluorocholine (^18^F-FCH) positron emission tomography–computed tomography (PET/CT). Nearly four years earlier the patient had been diagnosed in another hospital with metastatic prostatic adenocarcinoma (Gleason score 4 + 3, Ct2bN0M1 bone). The tumor was treated with external beam radiation therapy on the prostate bed and bone metastases (pubic bone and sacral lesion), and with androgen deprivation therapy and bisphosphonate. The disease was stabilized for nearly two years (according to the prostate specific antigen, PSA) and ^18^F-FCH levels taken as parameters. Despite the patient’s satisfactory clinical conditions, PSA was elevated to 937 ng/mL (previous value 82 ng/mL). The patient was started on enzalutamide 160 mg/day plus zoledronic acid 4 mg every 3 weeks, which resulted in a rapid decline of serum PSA levels (81 ng/mL). This biochemical response lasted for five months until PSA rose to 371 ng/mL. The patient was then prescribed 20 mg intra-venous docetaxel every three weeks in addition to 5 mg oral prednisone twice a day. Five months after the start of this regimen, the patient began to complain of a progressive and slowly worsening headache and diplopia. The patient’s Karnofsky performance scale (KPS) was 80. A restaging ^18^F-FCH PET/CT was scheduled and performed with a Siemens Biograph mCT nearly one hour after the intravenous administration of 163 MBq of ^18^F-FCH. The examination revealed an increased number of hypermetabolic lesions extensively involving the axial skeleton. Moreover, metabolic uptake was noted along the dura of the fronto-parietal convexities (transaxial PET, CT and fused PET/CT, Figure 1: **A** to **C**) in the left fronto-orbital region (transaxial PET, CT and fused PET/CT, Figure 1: **D** to **F**) and the right sphenoid region (transaxial PET, CT and fused PET/CT, Figure 1: **G** to **I**). These findings were strongly suggestive of meningeal metastatic implants. Since the seeding of prostate cancer metastatic cells in this location is highly unusual, a brain magnetic resonance examination was requested for a more detailed evaluation of these lesions. Unfortunately, the patient cannot be imaged due to claustrophobia.

**Figure 2 diagnostics-10-00385-f002:**
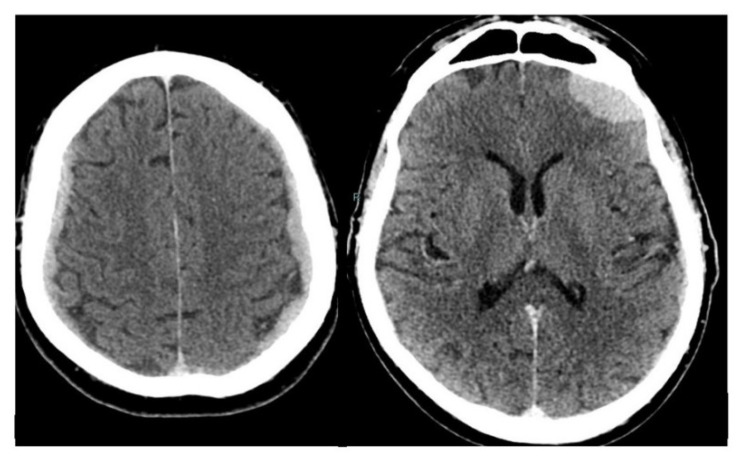
A brain contrast-enhanced computed tomography (CECT) scan was therefore performed and showed extensive pathologic iso-hyperdense tissue along the dura on both sides with intense and homogeneous enhancement following intravenous injection of the contrast medium. The dural-subdural pathologic tissue along the fronto-parietal convexities corresponded to the “en plaque” morphology discussed by Nzokou et al. [1] which may simulate sub-dural hematomas. In the weeks immediately following the CT scan, the patient’s clinical conditions rapidly worsened. He was treated with medical therapy with palliative intent. The patient died of arrhythmia approximately one month after the CT. Intracranial prostate cancer metastases are rare with an estimated range varying from 0.6% to 4.4% of cases [2]. In an extensive review of 16,280 patients with prostate cancer, only 28 (0.001%) were found to have dural metastases [3], while an autoptic study on 559 patients with hematogenous metastatic diffusion showed meningeal involvement in 5.9% of cases [4]. Since recent drug trials show improved overall survival for castration-resistant prostate cancer, it seems likely that cases of previously unrecognized dural metastases may become more clinically relevant. Cases of dural and parenchimal metastases from prostate cancer have been reported in the literature [5,6,7,8,9], but data on dural metastases with a bilateral “en plaque” morphological pattern are more limited [1,10]. To the best of our knowledge, no reports on ^18^F-FCH appearance in dural metastases from prostate cancer are present in the English scientific literature. As previously stated, the CT appearance of the metastatic lesions along the cerebral convexities was similar to sub-dural hematomas, but the pathologic tissue showed intense enhancement and was hypermetabolic on ^18^F-FCH PET/CT in contrast to hemorrhagic collections. Due to the extensive bilateral metastatic dural-subdural tissue, the patient was not considered eligible for surgery or whole-brain irradiation and received palliative care. In the present case, the use of combined molecular and morphological imaging avoided invasive diagnostic procedures in a patient with a dismal prognosis. The incidence of this unusual metastatic localization may be increasing as therapies for systemic cancer improve and patients survive longer.

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
