# Peer review of "Dural Metastases of Advanced Prostate Cancer Detected by 18F-Fluorocholine"

_diagnostics, 2020, doi:10.3390/diagnostics10060385_

Round 1
Reviewer 1 Report
I rode with interest the submission. It's just a clinical image, so no dramatic text changes are required. Nevertheless I recommend the following revisions:
- Please whenever possible insert a MRI image. This is a brain pathology, needs to be investigated with a contrast enhanced MRI. A clear statement justifying the unavailability of MRI imaging is clinically and scientifically completely ok, he could have had a pace-maker for example.
- The functional status of the patients is not clarified in a standard fashion: choose, either KPS or mRS and add it.
- A brief sentence expressing the reasons why the patient was (according to his/her functional status) not eligible for surgery: a growing amount of evidences suggests single until a maximum of three large (D>2.5 cm) intracranial mets could and should be surgically treated even in M1 patients, since brain metastases negatively impact the expected overall survival, independently from the extracranial spread of the disease.
- What was exactly the cause of death? should be added in the last sentence.
An English check is suggested.
Overall: Good work.
Author Response
- Q - Please whenever possible insert a MRI image. This is a brain pathology, needs to be investigated with a contrast enhanced MRI. A clear statement justifying the unavailability of MRI imaging is clinically and scientifically completely ok, he could have had a pace-maker for example.
- R – The patients was claustrophobic (view in text line 49)
- The functional status of the patients is not clarified in a standard fashion: choose, either KPS or mRS and add it. KPS was 80 (line 39)
- Q - A brief sentence expressing the reasons why the patient was (according to his/her functional status) not eligible for surgery: a growing amount of evidences suggests single until a maximum of three large (D>2.5 cm) intracranial mets could and should be surgically treated even in M1 patients, since brain metastases negatively impact the expected overall survival, independently from the extracranial spread of the disease.
- R – The patients refused any surgical treatement
- Q - What was exactly the cause of death? should be added in the last sentence
- R – Arrhythmia nearly one month after the examination.
An English check is suggested.
The text was edited by a PhD in English Literature

Reviewer 2 Report
Review: Dural Metastasis of Advanced Prostate Cancer Detected by 18 3 F-Fluorocholine
General Impression: This is an interesting case report which demonstrates dural metastasis detected by 18F-FCH PET/CT.
Although this is a case report following the structure of Introduction, Materials and Methods, Results/Discussion and Conclusion would be a good organizing principle.
In the introduction, it would be helpful to the readers, not as expert as the authors, why 18F-FCH was used instead of other PET CT imaging? Is it more sensitive and specific than FDG PET, for example?
The caption of Figure 1 could be separated into a methods section, with the pertinent findings in a Results section.
The discussion begins from line 52.
There are a few usage/grammatical problems which can be easily corrected:
line15: prostate cancer, but in relatively good clinical conditions, would better read: 15 prostate cancer, and in relatively good clinical condition.
line15: initially complaining of headache and diplopia. Would better read: The patient returned with complaints of headache and diplopia.
Line 17: lesions. Moreover, some suspicious dural-subdural.. delete Moreover
Line 18: suggestive of.. suggest changing to: concerning for
Line 20: performed to further clarify this picture and the images showed corresponding spontaneously
Line 21 hyperdense tissue at CT examination.
Recommend: performed which showed corresponding enhancing tissue which correlated with the PET findings.
Line 21: delete Thus,
Line 22 change to: prostate cancer, and associated with poor outcome.
Line 22: metastatization is awkward usage, recommend: metastasis.
Line 31 deprivation therapy and bisphosphonate. The disease thus stabilized for nearly two years…. recommend: The disease was stabilize for nearly two years..
Line 33: change raised to: was elevated
Line 34 zoledronic acid 4 mg every 3 weeks, thus resulting in a quick fall of serum
Change to: 34 zoledronic acid 4 mg every 3 weeks, which resulted in a rapid decline…
Line 35: The patient was hence prescribed would better read: The patient was then prescribed…
Line 40 and following recommend: Metabolic uptake was noted along the dura
Line 41: in the left fronto-orbital region (transaxial PET, CT and fused PET/CT, panels A to F,)
Line 42: and the right sphenoid region (transaxial PET, CT and fused PET/CT).
Line 61 recommend: In the weeks immediately following
Line 62: the CT scan, the patient’s clinical conditions rapidly worsened. He was treated with medical therapy
Line 63 with palliative intent. The patient died approximately one month after the CT. These lines should be in the “results” section. I suggest the article end with the conclusion:
Lines 60-61: In the present case, the use of combined molecular and 61 morphological imaging avoided invasive diagnostic procedures.
The authors could also supplement their references:
Andre Nzokou, Elsa Magro, François Guilbert, Jean Yves Fournier, Michel W. Bojanowski, Subdural Metastasis of Prostate Cancer
J Neurol Surg Rep. 2015 Jul; 76(1): e123–e127. Published online 2015 May 13. doi: 10.1055/s-0035-1549224
PMCID: PMC4520961
Vaios Hatzoglou, Gita V. Patel, Michael J. Morris, Kristen Curtis, Zhigang Zhang, Weiji Shi, Jason Huse, Marc Rosenblum, Andrei I. Holodny, Robert J. Young Brain Metastases from Prostate Cancer: An 11-Year Analysis in the MRI Era with Emphasis on Imaging Characteristics, Incidence, and Prognosis. J Neuroimaging. 2014 Mar-Apr; 24(2): 161–166.
Dural Metastases in Advanced Prostate Cancer: A Case Report and Review of the Literature A.B. Weiner, S. Cortes-Mateus, E. De Luis, I. Durán Curr Urol. 2014 Feb; 7(3): 166–168. Published online 2014 Feb 10. doi: 10.1159/000343558 PMCID: PMC4025047
Three cases of brain metastasis from castration‐resistant prostate cancer Yohei Shida, Tomoaki Hakariya, Yasuyoshi Miyata, Hideki SakaiClin Case Rep. 2020 Jan; 8(1): 96–99. Published online 2019 Dec 4. doi: 10.1002/ccr3.2587 PMCID: PMC6982515
Prostate carcinoma mimicking a sphenoid wing meningioma Lucas H. Bradley, Matthew Burton, Murat Gokden, Demitre Serletis Int J Surg Case Rep. 2015; 15: 63–65. Published online 2015 Aug 14. doi: 10.1016/j.ijscr.2015.08.018 PMCID: PMC4601959
Leptomeningeal Metastases in Hormone Refractory Prostate Cancer Jessica Y Ng, Justin Y Ng Cureus. 2018 Apr; 10(4): e2470. Published online 2018 Apr 12. doi: 10.7759/cureus.2470 PMCID: PMC5997428
Nayak L, Abrey LE, Iwamoto FM. Intracranial dural metastases. Cancer. 2009;115(9):1947‐1953. doi:10.1002/cncr.24203
Author Response
- Q - Although this is a case report following the structure of Introduction, Materials and Methods, Results/Discussion and Conclusion would be a good organizing principle.
- R -We followed the organizing scheme suggested by the editorial board for interesting images submission. We partly agree with the reviewer. However, a rearrangement of the whole of the structure should need a major rewriting of the text.
- Q - In the introduction, it would be helpful to the readers, not as expert as the authors, why 18F-FCH was used instead of other PET CT imaging? Is it more sensitive and specific than FDG PET, for example?
- R – Prostate cancers cell are characterized by an high turnover of the membrane phospholipids. 18F-FCH is a precursor of this metabolic pathway. We added a sentence in the abstract.
- Q - The caption of Figure 1 could be separated into a methods section, with the pertinent findings in a Results section.
- R - We followed the organizing scheme suggested by the editorial board for interesting images submission
Moreover all the other suggestions concerning the text have been added.

Reviewer 3 Report
The authors show that 18F-FCH PET-CT is useful for diagnosing dural metastasis. The manuscript is well written, but there are several concerns.
Major
- There is no description about “subdural hematomas” in the manuscript, though it is described in keywords. The authors need to explain the point of differentiation from subdural hematomas in the text.
- Panels A-C and panel D-F are similar, so it seems that one of them is unnecessary. If the authors have images of the sphenoid region on the right side, changing panel D-F to those would give a better impact.
- The authors describe that the mass shows an “en plaque” morphological pattern. The authors should explain the reason for using the word "en plaque" as plainly as possible, using the findings of contrast-enhanced CT and PET / CT.
Minor
- In References: The number “6” is missing.
Author Response
- Q - There is no description about “subdural hematomas” in the manuscript, though it is described in keywords. The authors need to explain the point of differentiation from subdural hematomas in the text.
- R – Added.
- Q -Panels A-C and panel D-F are similar, so it seems that one of them is unnecessary. If the authors have images of the sphenoid region on the right side, changing panel D-F to those would give a better impact.
- R - Done
- Q - The authors describe that the mass shows an “en plaque” morphological pattern. The authors should explain the reason for using the word "en plaque" as plainly as possible, using the findings of contrast-enhanced CT and PET / CT.
- R -Done.
Minor
- Q - In References: The number “6” is missing.
- R – Added.
